# Gene Expression Patterns in Lung Adenocarcinoma Cells in Response to Changes in Deuterium Concentration

**DOI:** 10.3390/ijms262210969

**Published:** 2025-11-12

**Authors:** Gabor I. Csonka, András Papp, Ildikó Somlyai, Gábor Somlyai

**Affiliations:** 1Department of Physics and Engineering Physics, Tulane University, New Orleans, LA 70118, USA; 2Department of Public Health, Faculty of Medicine, University of Szeged, H-6720 Szeged, Hungary; papp57andras@freemail.hu; 3HYD LLC for Cancer Research and Drug Development, H-1118 Budapest, Hungary; isomlyai@hyd.hu (I.S.); gsomlyai@hyd.hu (G.S.)

**Keywords:** deuterium concentration modulation, deuterium-depleted water, in vitro, lung adenocarcinoma, A549 cells, gene expression profiling, NanoString technology, oncogenic signaling, apoptosis regulation, drug resistance

## Abstract

Deuterium, a stable isotope of hydrogen present in natural water at ~150 ppm, has been implicated in modulating cellular metabolism and tumor progression. While deuterium-depleted water (DDW) has shown anti-cancer effects in preclinical and clinical studies, the underlying transcriptional mechanisms remain incompletely defined. Here, we profiled gene expression in A549 lung adenocarcinoma cells cultured for 72 h in media containing four graded deuterium concentrations (40, 80, 150, and 300 ppm) using a targeted NanoString panel of 236 cancer-related genes. After stringent quality filtering, 87 genes were retained and classified into nine distinct expression patterns based on fold-change trends relative to the 150 ppm control. High deuterium (300 ppm) induced strong upregulation (up to 2.1-fold) of oncogenic and survival-related genes (e.g., *EGFR*, *CTNNB1*, *STAT3*, *CD44*), while DDW (40–80 ppm) led to selective downregulation (down to 0.58-fold) of oncogenes (e.g., *MYCN*, *ETS2*, *IRF1*) and drug-resistance genes (e.g., *ABCB1*). Se-veral genes involved in DNA repair, apoptosis, and extracellular matrix remodeling exhibited dose-dependent responses, suggesting coordinated regulation by deuterium abundance. These findings demonstrate that deuterium concentration functions as a biologically active variable capable of modulating cancer-relevant gene networks. This exploratory dataset refines mechanistic models of DDW action and provides a foundation for future studies incorporating biological replication, functional assays, and in vivo validation. Significance: Deuterium concentration modulation alters oncogenic, apoptotic, and drug-resistance gene networks in lung adenocarcinoma cells, refining prior models of deuterium-depleted water effects. These findings identify deuterium concentration as a biologically active variable warranting further mechanistic and translational investigation.

## 1. Introduction

Cancer progression is driven by complex interactions between oncogenes, tumor suppressors, and environmental factors [1]. While the role of oncogenes and tumor suppressors is well established, the contribution of stable isotopes such as deuterium to tumor biology has received comparatively little attention. Naturally occurring at ~150 ppm in drinking water [2], deuterium is incorporated into biomolecules through metabolic processes, subtly altering reaction kinetics and hydrogen-bonding networks [3,4]. A recent study [5] provides a mechanistic framework for these observations, implicating quantum-level kinetic isotope effects—particularly those affecting hydrolytic enzyme reactions—as a potential driver of deuterium’s biological activity. The impact of deuterium substitution on hydrogen bonding and vibrational dynamics has also been well-characterized in model systems [6], where isotope effects influence tunneling, anharmonic motion, and molecular stability. Deuterium substitution can reduce enzymatic reaction rates by up to 6-fold due to kinetic isotope effects, primarily driven by differences in zero-point energy and bond strength compared to hydrogen [7]. These mechanisms may contribute to altered gene expression, cell-cycle progression, and the acquisition of drug-resistant phenotypes—hallmarks of malignant transformation and tumor progression. Given these links, modulation of deuterium abundance represents a potential metabolic and signaling checkpoint in tumor biology. Deuterium-depleted water (DDW) has been shown to inhibit proliferation in multiple cancer models. In human observational studies involving over 2600 patients with breast, lung, colorectal, pancreatic, prostate, ovarian, head and neck, brain, renal, bladder, skin, and hematologic cancers, DDW significantly prolonged survival when administered alongside conventional therapies—highlighting its potential translational relevance in clinical oncology [8]. Proposed mechanisms include modulation of mitochondrial function, redox balance, and signal transduction pathways [9]. By mo-dulating fundamental molecular interactions, shifts in the intracellular and extracellular D/H ratio may contribute to altered gene expression programs, changes in cell-cycle progression [10,11,12], and the acquisition of drug-resistant phenotypes—hallmarks of malignant transformation and tumor progression. Given these mechanistic links, modulation of deuterium abundance represents a potential metabolic and signaling checkpoint in tumor biology [13]. Previous studies have suggested that lowering the D/H ratio can influence mitochondrial function [14], oxidative stress responses, and apoptosis induction in malignant cells, while sparing normal tissues [15,16].

Two studies have investigated the effect of DDW on a limited set of cancer-related genes in carcinogen-induced mice. The findings showed that DDW inhibited the upregulation of *Kras*, *H-ras*, *Bcl2*, *Myc*, and *p53* gene expression, further supporting the role of an elevated D/H ratio in regulating gene expression [17,18]. However, the broader transcriptomic landscape underlying these effects remains to be better defined.

To address this gap, NanoString technology can be applied to profile [19] gene expression [20,21] for cancerous cells exposed to variable deuterium concentrations. Kovács et al. [22] previously profiled the expression of 236 cancer-related genes in A549 lung adenocarcinoma cells cultured under varying deuterium concentrations (40, 80, 150, and 300 ppm) using NanoString technology. Their analysis applied thresholds of ≥30% expression change and ≥30 mRNA copies. Under these criteria, 97.3% of cancer-related genes were upregulated at 300 ppm, while only five genes were downregulated under deuterium-depleted conditions. In parallel, comparative analysis of DDW’s impact on cell proliferation across human breast adenocarcinoma (MCF7), A549, and colorectal adenocarcinoma (HT29) cell lines showed that A549 cells exhibited the highest sensitivity [16], supporting their selection for the present study. The gene panel was curated to include genes relevant to proliferation, apoptosis, and drug resistance.

Building on this dataset, we applied a refined analytical pipeline designed to enhance measurement reliability and biological interpretability. As detailed in Section 4.5, we implemented stringent data cleaning criteria to exclude genes with high replicate variability or low expression, using coefficient of variation, robust Z-score and expression copy number thresholds tailored to each deuterium concentration. These filters reduced the dataset from 236 to 87 high-confidence genes, enabling classification of transcriptional responses into fivefold-change categories. To ensure that observed changes reflect true biological shifts rather than noise, all fold changes were accompanied by propagated error estimates derived from replicate variability as detailed in Section 4.6.

Using this error-controlled dataset, we systematically examined gene-specific responses to deuterium concentration changes. Expression ratios at 40 ppm, 80 ppm, and 300 ppm were compared to the 150 ppm control, revealing nine distinct transcriptional response patterns. These patterns—ranging from monotonic upregulation to biphasic or stable behavior—capture reproducible shifts in gene activity and provide a framework for biological interpretation. Pattern-based grouping revealed functional coherence, with several subsets. This classification enables pathway-level insights into how deuterium concentration modulates cancer-relevant transcriptional programs.

## 2. Results

### 2.1. Gene Expression Classification Results

As described above we selected 87 genes (see Section 4 for details) and classified them based on their expression ratios relative to the 150 ppm control. All fold-change values reported below were calculated from NanoString counts as described in Section 4.5. The summary of expression ratios in each classification category (upregulated, stable and downregulated) for the different deuterium levels is presented in Table 1.

To assess global transcriptional shifts across deuterium concentrations, we calculated average fold-change ratios and their associated dispersion metrics. At 40 ppm and 80 ppm, the mean downregulation ratios were 0.906 and 0.952, with corresponding mean absolute deviations of 0.096 and 0.063, respectively. At 300 ppm, the mean upregulation ratio was 1.471 with a deviation of 0.171. These values reflect the spread of fold-change responses across the 87 retained genes.

In parallel, we computed propagated errors for each individual fold-change ratio to quantify measurement uncertainty derived from replicate variability. The mean propagated errors were 0.150 (40 ppm), 0.106 (80 ppm), and 0.212 (300 ppm), indicating that expression measurements at high deuterium levels were more variable. Among all 261 fold-change measurements (87 genes × 3 conditions), *DAPK1* at 300 ppm exhibited the largest propagated error (±0.57) with strong upregulation (2.13-fold). Unlike mean absolute deviation, which summarizes dispersion across gene groups, propagated error captures the reliability of each fold-change value and is used to distinguish true biological shifts from noise.

Table 1 summarizes the distribution of gene expression responses across deuterium concentrations. At 300 ppm, no genes showed downregulation; instead, 81 genes were upregulated, with 35 classified as strong (>1.5-fold) and 46 as moderate (1.2–1.5-fold). This pattern contrasts sharply with the deuterium-depleted conditions, where 40 ppm medium led to 19 downregulated genes and only two moderately upregulated genes (1.23-fold), while 80 ppm medium produced 7 downregulated genes and no upregulation above the 1.2-fold threshold. Most genes at low deuterium concentrations remained within the stable range (66 at 40 ppm and 80 at 80 ppm). Notably, 10 of the 66 “stable” genes at 40 ppm had expression ratios below 0.85, placing them close to the moderately downregulated category.

### 2.2. Pattern Analysis and Gene Functions

To better understand the biological implications, the 87 retained genes were categorized into distinct expression patterns based on their behavior across the 40 ppm, 80 ppm, and 300 ppm deuterium concentrations as described in Section 4.6. Detailed data—including raw copy numbers, means, coefficients of variation (CVs), average copy number ratios relative to the 150 ppm reference, pattern assignments, and propagated errors—are provided in the Appendix A. Our pathway-level grouping aligns with the TCGA PanCanAtlas framework, which defines ten canonical oncogenic signaling pathways based on multi-omic analysis across 33 cancer types [23].

#### 2.2.1. Pattern 1: Stable at 40 ppm and 80 ppm, Strongly Upregulated at 300 ppm (29 Genes)

This pattern represents genes that maintain stable expression at lower deuterium concentrations (40 ppm and 80 ppm) but exhibit a significant increase in expression at high deuterium (300 ppm). These genes are enriched in oncogenic signaling, DNA repair, cell adhesion, and survival pathways. Their coordinated induction suggests activation of pro-proliferative and stress-response programs under high deuterium conditions (see Figure 1 legend for representative genes and propagated errors).

Genes in Pattern 1: *BRAF*, *BRCA1*, *CD44*, *CTNNB1*, *DAPK1*, *DEK*, *EGFR*, *ERBB3*, *ERCC4*, *FAT1*, *FGFR1*, *IFNGR1*, *ITGB1*, *JUN*, *LAMB1*, *LIF*, *MSH6*, *MYC*, *PLAUR*, *PTGS2*, *RB1*, *SPP1*, *STAT3*, *TFRC*, *TGFA*, *TGFBI*, *TGFBR2*, *XRCC5*, *YES1*.

Common Characteristics: These genes are predominantly involved in core oncogenic processes.

Cell Signaling Pathways: Many are components of key growth-promoting pathways such as MAPK/ERK (*BRAF*, *EGFR*, *JUN*, *YES1*), PI3K/AKT (*EGFR*, *ERBB3*, *LIF*, *MYC*), Wnt (*CTNNB1*), and JAK/STAT (*LIF*, *STAT3*). Their upregulation at 300 ppm suggests an amplification of these pro-survival and pro-proliferative signals under high deuterium conditions.Proliferation and Survival: Oncogenes like *MYC*, *EGFR*, and *ERBB3* directly promote cell cycle progression and inhibit apoptosis. *SPP1* and *TGFA* further enhance proliferation and tumor growth.Stress Response and DNA Repair: Genes like *BRCA1*, *ERCC4*, *MSH6*, and *XRCC5* are crucial for maintaining genomic stability through DNA repair mechanisms. Their upregulation might indicate a cellular response to deuterium-induced stress, where increased repair capacity is needed to cope with potential DNA damage. *DAPK1*, a pro-apoptotic gene, may also be activated in response to stress.Cell Adhesion and Migration: *CD44*, *ITGB1*, *LAMB1*, *FAT1*, *PLAUR*, and *TGFBI* are integral to cell–matrix interactions and extracellular matrix (ECM) remodeling, processes critical for tumor invasion and metastasis. Their strong upregulation at 300 ppm suggests an enhanced metastatic potential.Immune and Inflammatory Modulation: *IFNGR1* and *PTGS2* (COX-2) play roles in immune responses and inflammation, which can contribute to a tumor-supportive microenvironment.

Figure 2 illustrates Pattern 1 using the two most strongly upregulated genes, *DAPK1* and *TGFBR2*. Both exhibited pronounced induction at 300 ppm deuterium, indicating potential involvement in a coordinated regulatory response to shifts in the deuterium-to-hydrogen (D/H) ratio (see Figure 2 legend).

#### 2.2.2. Pattern 2: Stable at 40 ppm and 80 ppm, Moderately Upregulated at 300 ppm (32 Genes)

This pattern includes genes that are stable at lower deuterium levels but show a moderate increase in expression at 300 ppm.

Genes in Pattern 2: *AKT2*, *BCL2*, *BCL2L1*, *CAV1*, *CCND1*, *CCND3*, *CDK6*, *CLTC*, *CSK*, *E2F3*, *ERBB2*, *FAS*, *FOSL2*, *GAPDH*, *GRB7*, *HRAS*, *IGFBP2*, *MET*, *MST1R*, *NOTCH1*, *NUMA1*, *OGG1*, *PCNA*, *PDGFA*, *PIM1*, *RARA*, *STAT1*, *TGFB1*, *TOP2A*, *TUBB*, *TYMS*, *TYRO3*.

Common Characteristics:Cell Survival and Apoptosis Regulation: Genes like *BCL2*, *BCL2L1*, *AKT2*, and *PIM1* are anti-apoptotic, promoting cancer cell survival. *FAS*, a pro-apoptotic receptor, also falls into this category, suggesting a fine-tuned balance in cell death pathways.Cell Cycle Regulation: *CCND1*, *CCND3*, *CDK6*, *E2F3*, *PCNA*, and *TOP2A* are directly involved in cell cycle progression, particularly the G1–S transition and DNA replication. Their moderate upregulation at 300 ppm supports increased proliferative activity.Cell Proliferation and Growth: *ERBB2* (HER2), *HRAS*, *MET*, *MST1R*, and *PDGFA* are components of various growth factors signaling pathways, driving cell proliferation.Oncogenic Potential: Many genes in this pattern (e.g., *AKT2*, *BCL2*, *CCND1*, *ERBB2*, *HRAS*) are well-known oncogenes, frequently dysregulated in cancer. Their moderate upregulation at 300 ppm further supports the pro-cancer effects of high deuterium.DNA Repair and Maintenance: *OGG1*, *PCNA*, *TOP2A*, and *TYMS* are crucial for DNA repair and synthesis, ensuring genomic stability necessary for rapid cancer cell proliferation.

Connections: These genes are integral to major oncogenic pathways (*PI3K/AKT, RAS/MAPK, TGF-beta, NOTCH*). Their stability at 40 and 80 ppm and moderate upregulation at 300 ppm suggest a sensitivity to deuterium levels, possibly due to subtle effects on enzyme kinetics or membrane fluidity, which can be altered by deuterium’s heavier mass [27].

#### 2.2.3. Pattern 3: Moderately Downregulated at 40 ppm, Stable at 80 ppm, Upregulated at 300 ppm (15 Genes)

This pattern characterizes genes that show moderate suppression at 40 ppm, stability at 80 ppm. Of the 15 genes in Pattern 3, 3 were strongly upregulated and 12 moderately at 300 ppm.

Genes in Pattern 3: *ETS2*, *HIF1A*, *HSP90AB1*, *IRF1*, *LYN*, *MLL (KMT2A)*, *MMP9*, *NPM1*, *PTPRG*, *RAF1*, *RET*, *RRM1*, *SERPINE1* (PAI-1), *TPR*, *XPC*.

Six of the Pattern 1 genes: *BRAF*, *CD44*, *CTNNB1*, *DEK*, *EGFR*, *ITGB1* show downregulation between 0.83 and 0.85 in 40 ppm medium thus these genes are close to the border between pattern 1 and 3. Two of the Pattern 2 genes *TOP2A* and *TYMS* show similar downregulation.

Common Characteristics:Cancer Progression and Survival: Most genes in this pattern are associated with hallmarks of cancer, including cell invasion (*MMP9*), proliferation (*RAF1*, *RET*), angiogenesis (*HIF1A*, *SERPINE1*), and cell survival (*HSP90AB1*, *LYN*).Deuterium Sensitivity: The moderate downregulation observed at 40 ppm and the upregulation at 300 ppm suggest that the expression of these genes is bidirectionally responsive to shifts in deuterium concentration. While these genes may not be directly involved in classical signaling pathways, their functions are nonetheless fundamental, indicating that they play a critical role in maintaining cellular homeostasis under varying D/H ratios.Diverse Cellular Processes: These genes span a wide range of cellular functions, including cell adhesion, transcription, signaling, matrix remodeling, DNA repair, and protein stability. This diversity indicates a broad network of cancer-related functions that are collectively modulated by deuterium’s effects on cellular redox state.

Connections: These genes form a highly interconnected network involving MAPK, hypoxia, and invasion pathways.

#### 2.2.4. Pattern 4: Strongly Downregulated at 40 ppm, Stable at 80 ppm, Upregulated at 300 ppm (2 Genes)

This pattern describes genes that are significantly suppressed at very low deuterium (40 ppm), return to stable levels at 80 ppm, and become moderately or strongly upregulated at high deuterium (300 ppm). This is an interesting patten as the variable deuterium concentration has a large, steady effect on the copy numbers, the expression of the gene.

Genes in Pattern 4: *BCL2A1*, *TGFBR3*.

Figure 3 shows the pattern for *BCL2A1*, *TGFBR3* genes. The *BCL2A1* gene expression is rapidly increasing in the range of 40–80 ppm deuterium concentrations, in the next concentration range it is stable and in the 300 ppm concentration medium it abruptly increases to be very strongly upregulated. *TGFBR3* expression shows a steady increase from strong down regulation to moderate upregulation.

#### 2.2.5. Pattern 5: Stable at All Concentrations (4 Genes)

This pattern identifies genes whose expression remains stable across all tested deuterium concentrations (40 ppm, 80 ppm, 150 ppm, and 300 ppm). In the 40 ppm medium, expression levels for these genes decrease to 84–89%, positioning them just above the 83% cutoff for moderate downregulation.

Genes in Pattern 5: *FANCG*, *MUC1*, *PCTK1*, *TFE3*.

Common Characteristics:Cellular Protection and Maintenance: *FANCG* is involved in DNA repair, ensuring genomic integrity. *MUC1* provides mucosal protection and can play roles in cell survival signaling. *TFE3* regulates lysosomal biogenesis and metabolic homeostasis. *PCTK1* supports cell cycle progression and neuronal differentiation.Limited Oncogenic Roles: While *MUC1* and *TFE3* have known associations with cancer, *FANCG* and *PCTK1* are not primarily considered oncogenic drivers in the same vein as genes in Patterns 1 and 2.Stability Across DDW Levels: The consistent stability of these genes across all deuterium concentrations suggests that their functions are essential and tightly regulated, making them robust against variations in deuterium levels. This may indicate that these processes are fundamental for cell survival and are not significantly mo-dulated by deuterium, or that their regulatory mechanisms are insensitive to these specific deuterium shifts.

Connections: These genes primarily function in cellular protection, maintenance, and basic cellular processes. Their stability contrasts with the dynamic regulation observed in other patterns, highlighting a subset of genes whose expressions show small dependence on deuterium concentration within the tested range.

#### 2.2.6. Pattern 6: Moderately Downregulated at 40 ppm and 80 ppm, Stable at 300 ppm (1 Gene)

This pattern identifies a gene that is moderately suppressed at both 40 ppm and 80 ppm deuterium but shows stable expression at 300 ppm.

Gene in Pattern 6: *FGFR4*.

Characteristics:Oncogenic Receptor Tyrosine Kinase: *FGFR4* is a receptor tyrosine kinase involved in cell proliferation, differentiation, and survival, often overexpressed in various cancers (e.g., hepatocellular carcinoma).Deuterium Sensitivity: The moderate downregulation at 40 ppm and 80 ppm suggests that lower deuterium levels may suppress *FGFR4* expression. Its stability at 300 ppm, unlike the upregulation seen in many other oncogenes at high deuterium, might indicate tight regulatory control or a saturation of its signaling at normal/high deuterium levels.

The suppression of *FGFR4* at low deuterium supports the potential anti-cancer effects of DDW by inhibiting oncogenic signaling.

#### 2.2.7. Pattern 7: Stable at 40 ppm, Strongly Downregulated at 80 ppm, Moderately Upregulated at 300 ppm (1 Gene)

This unique pattern includes a single gene, *ABCB1*, which is shown in Figure 4.

#### 2.2.8. Pattern 8. Moderately Upregulated at 40 ppm, Stable at 80 ppm, Upregulated at 300 ppm (2 Genes)

This unique pattern contains genes that are moderately upregulated at 40 ppm deuterium concentration to 1.23 ± 0.04. No other gene shows such a considerable upregulation. *BCL3* is moderately (almost strongly) upregulated to 1.49 ± 0.32, while *PTK7* is strongly upregulated to 1.66 ± 0.12 at 300 ppm deuterium (cf. Appendix A).

Genes in Pattern 8: *BCL3* and *PTK7*.

Common Characteristics:Oncogenic and Pro-Metastatic Roles: *BCL3* is an oncogenic transcription co-regulator promoting cell proliferation and survival. *PTK7* is a pseudokinase involved in Wnt signaling, cell polarity, and migration, often promoting invasion and metastasis.Unique Deuterium Sensitivity: The moderate upregulation at 40 ppm is distinct from most other patterns, suggesting a unique sensitivity to very low deuterium. Their upregulation at 300 ppm aligns with general cancer gene activation at high deuterium.

Connections: This pattern suggests that *BCL3* and *PTK7* may enhance cancer progression under varying deuterium levels, potentially complicating the anti-cancer effects of DDW. Their regulation appears to be sensitive to subtle shifts in cellular environment induced by deuterium.

#### 2.2.9. Pattern 9: Stable at 40 ppm, Moderately Downregulated at 80 ppm, Stable at 300 ppm (1 Gene)

At 40 ppm, MYCN expression shows a fold change of 0.89 ± 0.11 relative to the 150 ppm control, indicating a slight decrease and considered stable. At 80 ppm, expression drops to 0.76 ± 0.01, reflecting moderate downregulation with high confidence. At 300 ppm, expression is 1.01 ± 0.15, consistent with baseline and interpreted as stable.

Characteristics:Potent Proto-Oncogene: *MYCN* is a transcription factor critical for neural development but is frequently amplified or overexpressed in aggressive cancers like neuroblastoma, driving rapid tumor growth.Complex Deuterium Response: Its probable stability at 40 ppm and 300 ppm suggests tight regulatory control or context-specific roles that prevent its activation by extreme deuterium levels. However, the moderate downregulation at 80 ppm indicates that moderate deuterium depletion can suppress *MYCN*, potentially inhibiting tumor growth, which aligns with DDW’s anti-cancer effects in neuroblastoma models.

This unique pattern reveals a nuanced response of *MYCN* to deuterium levels, suggesting that its suppression may depend on an optimal concentration range, while both extreme depletion and enrichment appear functionally neutral.

## 3. Discussion

The comprehensive analysis of gene expression in A549 lung cancer cells under varying deuterium concentrations reveals profound and diverse impacts on cancer-related pathways. The initial data cleaning process was crucial, demonstrating that a significant proportion of raw gene expression measurements are unreliable due to low copy numbers or high variability. The retention of 87 high-quality gene expressions provides a robust foundation for drawing biologically meaningful conclusions. While the dataset provides compelling insights into dose-dependent transcriptional and translational responses, it is important to note that the study was conducted using technical duplicates. Due to the limited replication, formal statistical testing (e.g., *p*-values, ANOVA) was not applied. Instead, we relied on fold-change thresholds, CV filtering to ensure robustness in gene selection and pattern classification (see Section 4.5 and Section 4.6). This metric provides a transparent framework for interpreting gene-specific responses and identifying biologically meaningful trends despite the absence of inferential statistics.

The most general observation is the strong and widespread upregulation of oncogenic and metastatic genes at 300 ppm deuterium. Genes like *EGFR*, *CTNNB1*, *RAF1*, *STAT3*, and *ITGB1*, which are central to cell proliferation, survival, and invasion, showed significant increases in expression. This aligns with existing literature suggesting that higher deuterium concentrations stimulate cell growth [9]. The coordinated upregulation of genes involved in MAPK/ERK, PI3K/AKT, Wnt, and JAK/STAT signaling pathways indicates that high deuterium may broadly stimulate pro-cancer signaling networks, potentially by influencing metabolic processes or inducing oxidative stress [29].

Conversely, lower deuterium levels (40 ppm and 80 ppm) generally led to a more stable expression profile for many essential cellular genes (Pattern 5: *FANCG*, *MUC1*, *PCTK1*, *TFE3*), suggesting that fundamental cellular processes like DNA repair and metabolic homeostasis are robust against deuterium variations. However, specific genes, particularly those involved in cell survival and drug resistance, exhibited interesting patterns of downregulation at lower deuterium levels. For instance, *ABCB1* (Pattern 7), a key multidrug resistance transporter, was downregulated at 80 ppm, implying that moderate deuterium depletion could potentially sensitize cancer cells to chemotherapy. Similarly, *MYCN* (Pattern 9), a potent proto-oncogene, was moderately downregulated at 80 ppm, suggesting an inhibitory effect of DDW on its expression.

The observed discrepancies and higher variability at 40 ppm deuterium warrant particular attention. While some of this variability could be attributed to technical factors, it is also plausible that extreme deuterium depletion induces a significant biological stress response in A549 cells. Genes like *AREG* (not retained in the cleaned set due to large CV and Z), which showed high CVs at 40 ppm, might represent genuine, albeit highly variable, biological responses to this extreme condition. Deuterium depletion can disrupt mitochondrial function and alter cellular redox states, potentially leading to oxidative stress or changes in enzyme kinetics that disproportionately affect certain sensitive genes [16]. The unique upregulation of *BCL3* and *PTK7* (Pattern 8) at 40 ppm further supports the idea of complex, gene-specific responses to very low deuterium, possibly through transient activation of NF-κB or other stress-responsive pathways.

Our novel classification system, dividing gene expression responses into nine distinct patterns (Pattern 1–9), elucidates the nuanced impact of deuterium concentration. Genes exhibiting upregulation at 300 ppm (Patterns 1, 2 and 3) were identified as key drivers of cancer progression. Consequently, targeted inhibition of these genes (e.g., *EGFR* with specific inhibitors) represents a plausible therapeutic strategy to counteract the pro-oncogenic effects associated with elevated deuterium levels. The pronounced, dose-dependent regulation of *ABCB1* and other ABC transporters is particularly noteworthy. These membrane proteins influence drug efflux and chemoresistance, and their modulation by deuterium content raises the possibility that DDW could sensitize tumors to chemotherapy by altering transporter activity. This aligns with emerging evidence that metabolic and isotopic environments can influence drug response.

These findings corroborate previous observations that DDW inhibits cell proliferation, whereas DEW stimulates it. This supports the hypothesis that an elevated intracellular D/H ratio is critical for initiating the G1–S phase transition [30]. A gene-specific analysis demonstrates that central cell cycle regulatory genes are activated under higher D/H conditions.

The observation that relatively few, 7 or 19 out of 87 genes less than 0.83-fold expression changes in response to DDW (cf. Table 1) suggests the primary regulatory mechanism of cell division may involve preventing an increase in the D/H ratio, rather than reducing it outright. This molecular mechanism underpins the observed effects of DDW in human applications: a decrease in systemic deuterium concentration impedes the activation of genes essential for cell proliferation. Furthermore, DDW exerts a systemic metabolic effect that induces cellular stress, leading to a marked increase in experimentally confirmed ROS levels and triggering apoptotic pathways [16].

Although no direct data currently demonstrate that Na^+^/H^+^ exchange alters intracellular D/H ratio, the hypothesis remains plausible given the exchanger’s role in modulating intracellular pH and ionic composition. Prior studies demonstrated the sensitivity of membrane transport processes to deuterium concentration. In a human phase II clinical trial, a significant reduction in blood serum Na^+^ concentration was observed after 90 days of consuming deuterium-depleted water (DDW, 105 ppm), decreasing from 141.1 mmol/L to 139.0 mmol/L (*p* = 0.000024) [31]. This finding suggests that deuterium concentration may influence ion transport mechanisms. Similarly, in a plant-based experiment, *Elodea canadensis* exposed to a medium with reduced deuterium content (150 ppm to 87 ppm) exhibited detectable external acidification within 30 min, indicating rapid membrane transport responses to D-level changes [32].

Mitochondrial involvement in D/H regulation is further supported by a murine study in which dietary supplementation with deuterium-depleted egg yolk prolonged survival following transplantation with the 4T1 mammary carcinoma cell line [33]. This effect is modulated by macronutrient composition: higher fat intake yields metabolically produced DDW, lowering intracellular deuterium levels and counteracting D/H elevation. In healthy cells, membrane transport and mitochondrial metabolism operate in concert to maintain D/H homeostasis. In contrast, cancer cells often exhibit dysfunctional TCA cycle activity [29], disrupting this balance and permitting deuterium accumulation. This shift may contribute to uncontrolled proliferation, as recent transcriptomic analyses confirm that elevated D/H ratios activate genes essential for cell growth [10].

In this experiment, the D concentration was adjusted to levels 70 and 110 ppm lower, and 150 ppm higher than the natural D concentration. This study demonstrated that variations in D concentration can affect the expression of cancer-related genes. This concept builds on an earlier study that examined the effects of D concentration on blood glucose levels in diabetic rats [34]. That initial study found that a D concentration of 25 ppm significantly reduced blood sugar levels. In a follow-up study testing seven D concentrations ranging from 75 to 150 ppm, researchers discovered that the optimal range for activating the insulin signaling pathway was between 125 and 140 ppm. These findings suggest that altering the D/H ratio—by increasing or decreasing the D concentration by approximately 20–30 ppm relative to natural levels—may be critical for modulating biological pathways supporting by a long-term observational study involving 2649 DDW-consuming patients [8].

Our transcriptomic survey of A549 lung adenocarcinoma cells across controlled deuterium concentrations reveals coordinated, dose-dependent modulation of cancer-relevant gene networks. Although the study is limited to technical replicates and a single cell line, the observed expression patterns refine earlier hypotheses regarding deuterium’s influence on oncogenic signaling, apoptosis regulation, and drug resistance pathways. The exclusive use of A549 lung adenocarcinoma cells provided a well-controllable model system, but it also represents a limitation. Similar transcriptional effects of deuterium modulation have been reported in other cancer models, suggesting that the responses described here may not be unique to lung adenocarcinoma. Nonetheless, validation in additional cancer cell types will be required to establish the broader relevance of these findings. The current findings establish deuterium concentration as a biologically active variable with potential relevance for cancer biology. Future investigations should incorporate biological replication, functional validation, and in vivo models to determine the translational significance of these molecular responses and their therapeutic implications.

## 4. Materials and Methods

### 4.1. Experimental Setup and Cell Culture

A549 lung cancer cells were cultured in water media with four different deuterium concentrations: 40 ppm, 80 ppm, 150 ppm (control), and 300 ppm. To prepare the media stock solutions were made up, each containing 6.68 g DMEM, 0.11 g NaHCO_3_, 0.01 g Streptomycin and 500 µL Penicillin, dissolved in either distilled water (150 ppm), DDW (40 and 80 ppm) or DEW (300 ppm) D concentration. These solutions were sterile filtered (Acrodisc Premium 25 mm Syringe Filter with 0.2 µm GHP Membrane, Pall Life Sciences, Port Washington, NY, USA, cat. P NAP-4564T). A549 cells were seeded into 10 cm diameter cell culture dishes 16 h prior to treatment, then incubated at 37 °C in a 5% CO_2_ atmosphere in serum-free medium. A total of 400,000 cells were seeded per dish. After 18 h, the medium was removed and replaced with media containing different concentrations of deuterium, which also included 10% fetal bovine serum (Gibco, Life Technologies Hungary Ltd., Budapest, Hungary). The cells were then incubated for 72 h. During incubation, cell proliferation was monitored daily in the culture dishes, as the incubation was intended to be terminated at approximately 60% confluency. Two parallel experiments were conducted for each treatment condition. At the end of the treatment, the medium was removed, and total RNA was isolated from the cells.

#### 4.1.1. Production of Deuterium-Depleted Water

DDW with a final deuterium concentration of 25 ppm was produced by HYD LLC for Cancer Research and Drug Development (Budapest, Hungary) by fractional distillation of purified ordinary tap water, which initially contained 150 ppm of deuterium (equivalent to 0.0158 mol/L HDO). Production followed Good Manufacturing Practice (GMP) standards for quality assurance. The final deuterium concentration was precisely verified using a Liquid-Water Isotope Analyser-24d (manufactured by Los Gatos Research Inc., San Jose, CA, USA) with a precision of ±1 ppm. For gene expression studies, DDW at 40 ppm and 80 ppm deuterium concentrations were used, alongside a 150 ppm deuterium concentration as a control. Additionally, deuterium enhanced water (DEW) at 300 ppm was prepared by adding 99.9% heavy water (obtained from Sigma Aldrich, Budapest, Hungary) to achieve the desired concentration. The different deuterium water sources were checked for batch-to-batch variability. The reaction H_2_O + D_2_O ⇌ 2 HDO, occurs on a picosecond timescale, thus the equilibrium is reached immediately, leading to a mixture of 99.4% H_2_O and 0.06% HDO and negligible D_2_O after thorough mixing.

#### 4.1.2. Cell Culture and Deuterium Exposure

A549 cells were seeded at a density of 400,000 cells per well in serum-free medium and incubated for 16 h prior to experimental manipulation. Following this initial incubation, the serum-free medium was aspirated and replaced with experimental media containing varying concentrations of deuterium (40 ppm, 80 ppm, 150 ppm, and 300 ppm), supplemented with fetal calf serum. The culture media were prepared to maintain consistent osmolarity and pH. All observed changes in cell growth and gene expression were attributed solely to differences in D concentration. Cells were cultured in these media for 72 h. All experimental groups, including a control group, were maintained under identical conditions. Cells were harvested when they reached approximately 60% confluence.

### 4.2. Gene Expression Quantification

NanoString technology was employed to quantify the expression of 236 cancer-related genes. NanoString’s nCounter Analysis System allows for direct, digital quantification of RNA molecules without the need for reverse transcription or amplification, minimizing potential biases. The technology utilizes unique color-coded molecular barcodes that hybridize to specific RNA targets, which are then counted by an nCounter Analysis System from NanoString digital analyzer, providing precise measurements of gene expression levels.

### 4.3. NanoString Data Analysis Pipeline

Raw RCC files from the nCounter system were processed using NanoString’s standard workflow. Background correction was performed using negative control probes, followed by positive control (CodeCount) normalization. Lane content normalization was applied using the geometric mean of all expressed targets, and final scaling was performed with the geometric mean of stably expressed housekeeping genes (e.g., *GAPDH*, *HPRT1*).

### 4.4. Dataset Description

The raw dataset comprised expression measurements for 236 cancer-related genes, with each gene measured in duplicate at each of the four deuterium concentrations. The correlation coefficient between the duplicate measurements is 0.9996, proving that the results are reproducible.

### 4.5. Data Cleaning Criteria

We used the Coefficient of Variation (CV) to find statistically relevant data. For pairwise comparisons of gene expression ratios, we quantified variability using a dimensionless coefficient of variation defined as CV = (x_1_ − x_2_)/(x_1_ + x_2_). This approach avoids inflated variance estimates from degrees-of-freedom corrections and provides a biologically interpretable measure of dispersion, especially under low-replicate conditions. The robust Z-score applied here helps identify outliers in gene expression differences while being less sensitive to extreme values than a traditional Z-score [35].

To ensure the reliability of downstream analyses, stringent data cleaning criteria were applied:High Coefficient of Variation Exclusion: Genes were excluded if their CV exceeded 20% and |Z| > 3 (a standard choice) at the 80 and 150 ppm deuterium concentration. These thresholds were selected to align with a moderate overexpression limit of 1.2-fold, ensuring that CVs remained within a range that supports reliable categorization. The 23% cutoff reflects a +3% adjustment to accommodate higher baseline variability at 40 and 300 ppm, preserving biologically relevant genes. CV distribution analysis revealed gaps at 20% (80 and 150 ppm) and 23% (40 and 300 ppm), further supporting these thresholds. High CV and Z indicate excessive replicate variability, likely due to experimental noise or biological heterogeneity that could confound interpretation.Low Average Copy Number Exclusion: Genes were excluded if their average copy number was less than 20 at the 150 ppm deuterium level. This filter targets low-expression genes, which are inherently more susceptible to technical noise and may lack significant biological relevance in this context.

After data cleaning, 87 genes were retained and categorized based on their expression ratios relative to a 150 ppm control. This reduction in the number of genes from 236 to 87 highlights the presence of significant noise and variability in the raw measurements.

### 4.6. Gene Expression Classification

Gene expression changes were classified using the following thresholds:Strongly Upregulated: Ratio > 1.5 (expression is more than 50% higher than in the control medium).Moderately Upregulated: 1.2 < Ratio ≤ 1.5 (expression is 20% to 50% higher).Stable: 0.83 ≤ Ratio ≤ 1.2 (expression is within the range of −17% to +20% of the control).Moderately Downregulated: 0.67 ≤ Ratio < 0.83 (expression is 17% to 33% lower).Strongly Downregulated: Ratio < 0.67 (expression is less than two-thirds of the control).

These thresholds combine a standard 1.5-fold cutoff for strong expression changes with a more sensitive 1.2-fold cutoff to capture subtle but potentially meaningful shifts—particularly for dosage-sensitive or regulatory genes where modest changes may have amplified effects. To ensure such changes exceed technical noise, we aligned the CV threshold (~20%) with the 1.2-fold limit, enabling reliable categorization of expression shifts near this boundary.

To ensure that observed differences were not artifacts of measurement variability, fold-change values were accompanied by propagated errors, calculated by combining the standard errors of the mean NanoString counts at each deuterium concentration and at the 150 ppm control. Specifically, the relative error of a ratio was derived using standard error propagation rules, where the squared relative errors of numerator and denominator were summed and the square root taken. This approach provides a conservative estimate of uncertainty around each fold-change, allowing us to distinguish genuine expression shifts from fluctuations within the error range.

### 4.7. Data Cleaning Outcomes

Retained Genes: Genes such as *BRCA1*, *TP53*, and *EGFR*, known for their critical roles as tumor suppressors or oncogenes, were consistently retained. These genes exhibited stable expression (CV < 10% across deuterium levels), indicating reliable measurements suitable for studying deuterium effects. The results are summarized in Appendix A.Excluded Genes: A significant number of genes were excluded due to either low expression (e.g., *AKT1*, with an average copy number of 4 at 40 ppm) or high variability (e.g., *AREG*, with a CV of 33.33% at 40 ppm). While the exclusion of low-expression genes helps reduce technical noise, the removal of high-variability genes warrants careful consideration as they might represent genuine biological responses to deuterium-induced stress rather than mere experimental artifacts. Notably, the 40 ppm deuterium condition exhibited the largest discrepancies and highest average CV (~18% compared to ~10–14% for other levels), would have contributed disproportionately to gene exclusions. This suggests that extreme deuterium depletion may induce greater biological variability or be more susceptible to technical inconsistencies. These problems will be investigated in future studies.

The NanoString experiment was repeated twice for four retained cancer genes *BRAF*, *CLTC*, *LYN* and *RAF1* in an independent experiment. For *BRAF* the copy numbers in the new experiment show a systematic increase of 100–105, but the pattern does not change (see pattern 1). The two experiments for *CLTC*, *LYN* show excellent agreement, while for *LYN* a negligible systematic decrease of 2–12 can be observed which again does not change the outcome of the detailed analysis.

## Figures and Tables

**Figure 1 ijms-26-10969-f001:**
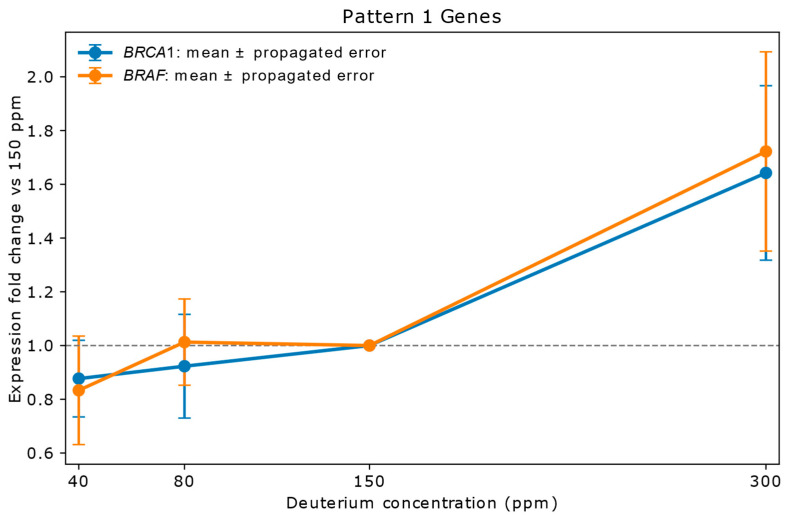
Relative mRNA expression of two representative Pattern 1 genes (*BRAF*, *BRCA1*) in A549 lung adenocarcinoma cells cultured for 72 h in media containing 40 ppm, 80 ppm, 150 ppm (natural abundance control), or 300 ppm deuterium. Expression values (*y*-axis) are shown as fold change relative to the 150 ppm control (*x*-axis), calculated from mean NanoString copy numbers. Vertical error bars represent propagated errors derived from replicate variability using standard error propagation rules. Pattern 1 genes are defined by stable expression at 40 ppm and 80 ppm (ratios within 0.83–1.20) and strong upregulation (>1.5-fold) at 300 ppm. The 300 ppm concentration triggered coordinated upregulation across multiple genes, suggesting activation of shared regulatory networks, including PI3K/AKT, p53, and NF-κB. D-induced changes in H-bonding may alter chromatin accessibility, influencing transcription rates. This pattern aligns with literature suggesting that higher deuterium concentrations can enhance oncogenic pathways [15].

**Figure 2 ijms-26-10969-f002:**
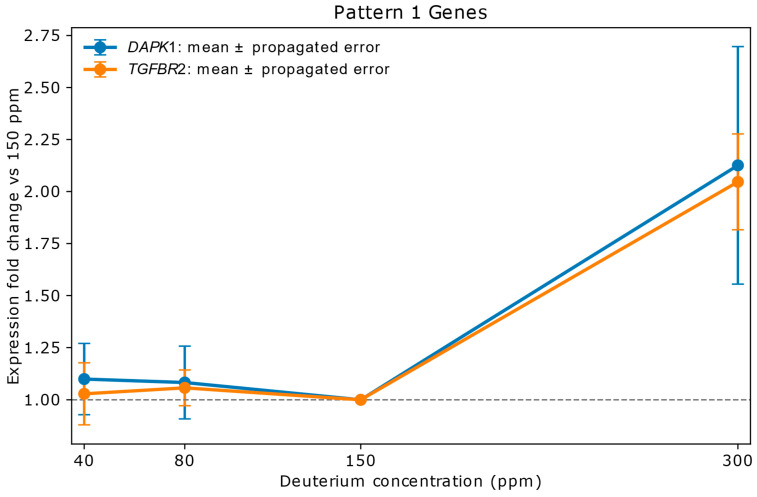
Relative mRNA expression of *DAPK1* and *TGFBR2* in A549 lung adenocarcinoma cells exposed to varying deuterium concentrations. Cells were cultured for 72 h in media containing 40 ppm, 80 ppm, 150 ppm (natural abundance control), or 300 ppm deuterium. Expression values (*y*-axis) are shown as fold change relative to the 150 ppm control (*x*-axis), with vertical bars indicating propagated errors from replicate variability. Both genes remained stable at low deuterium (40–80 ppm), with values falling within the propagated error range. At 300 ppm, expression was markedly upregulated: *DAPK1* increased 2.13-fold with an error of ±0.57, and *TGFBR2* rose 2.05-fold with an error of ±0.23. *TGFBR2* functions as a receptor for external signals—most notably TGF-β—while *DAPK1* serves as an intracellular effector involved in apoptosis and autophagy. Their potential intersection lies in TGF-β-induced cell death or cell cycle arrest: *TGFBR2* initiates the signaling cascade, and *DAPK1* may act downstream to execute the death signal, particularly under stress conditions such as elevated reactive oxygen species (ROS) levels [16,24,25,26].

**Figure 3 ijms-26-10969-f003:**
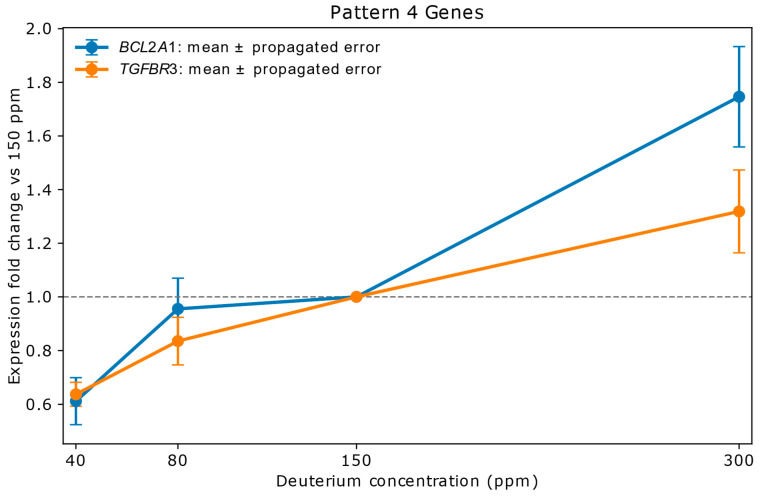
Relative mRNA expression of *BCL2A1* and *TGFBR3* in A549 lung adenocarcinoma cells cultured for 72 h in media containing 40 ppm, 80 ppm, 150 ppm (control), or 300 ppm deuterium. Expression values (*y*-axis) are shown as fold change relative to the 150 ppm control (*x*-axis), based on mean NanoString copy numbers. Vertical error bars represent propagated errors derived from replicate variability using standard error propagation rules. *BCL2A1*, an anti-apoptotic mitochondrial membrane protein, is strongly downregulated at 40 ppm, returns to stable levels at 80 ppm, and is sharply upregulated at 300 ppm. *TGFBR3*, a co-receptor modulating TGF-β signaling, shows a gradual increase from strong downregulation at 40 ppm to moderate upregulation at 300 ppm. These responses suggest that extreme deuterium depletion suppresses pro-survival signaling, while enrichment promotes it—a dynamic potentially exploitable in therapeutic modulation of apoptosis and metastasis. *BCL2A1* contributes to cancer cell survival under oxidative stress and is associated with chemotherapy resistance. *TGFBR3* may act as a tumor suppressor in early stages but promote metastasis in advanced cancers. The distinct expression trajectories of these genes highlight their sensitivity to deuterium concentration and underscore their roles in stress response and cell fate regulation.

**Figure 4 ijms-26-10969-f004:**
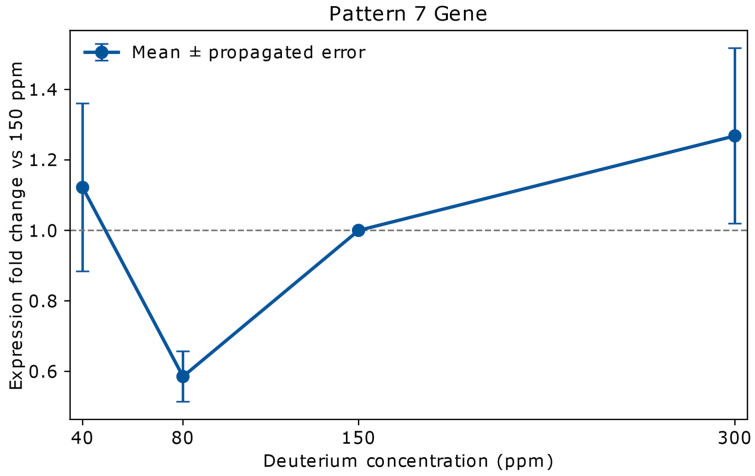
Relative mRNA expression of *ABCB1* in A549 lung adenocarcinoma cells cultured for 72 h in media containing 40 ppm, 80 ppm, 150 ppm (natural abundance control), or 300 ppm deuterium. Expression values (*y*-axis) are shown as fold change relative to the 150 ppm control (*x*-axis), based on mean NanoString copy numbers. Vertical error bars represent propagated errors derived from replicate variability using standard error propagation rules. *ABCB1*, a member of the ATP-binding cassette (ABC) transporter superfamily, encodes P-glycoprotein—an ATP-powered efflux pump that actively exports a broad range of hydrophobic molecules, including chemotherapeutic agents such as Vinca alkaloids, Taxanes, and Anthracyclines. At 40 ppm, expression of *ABCB1* shows a fold change of 1.12 ± 0.23, indicating a slight increase that falls within the propagated error margin and is therefore considered statistically stable. At 80 ppm, expression drops sharply to 0.58 ± 0.07, representing strong downregulation with high confidence. At 300 ppm, expression increases to 1.27 ± 0.24, consistent with moderate upregulation or stability. This dose-dependent pattern suggests that intermediate deuterium depletion may transiently suppress *ABCB1*-mediated drug efflux, potentially enhancing chemotherapy efficacy, whereas high deuterium could promote efflux-mediated resistance. The sensitivity of *ABCB1* to 80 ppm deuterium highlights a narrow therapeutic window for modulating multidrug resistance. Overexpression of *ABCB1* is a well-established mechanism by which cancer cells evade chemotherapy [28], making its regulation by deuterium concentration a potential target for therapeutic intervention. However, the large propagated errors at 40 and 300 ppm warrant further measurements.

**Table 1 ijms-26-10969-t001:** Summary of Expression Ratios of the 87 Retained Genes Relative to 150 ppm Control in Each Classification Category by Deuterium Content.

Deuterium Content	Strong Up (>1.5)	Moderate Up (1.2–1.5)	Stable (0.83–1.2)	Moderate Down (0.67–0.83)	Strong Down (<0.67)
40 ppm	0	2	66	17	2
80 ppm	0	0	80	6	1
300 ppm	35	46	6	0	0

## Data Availability

All gene expression data used in this study were obtained from previously published sources. Specific dataset identifiers and accession numbers are provided in the Appendix A. No new data were generated.

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
