# Peer review of "Gene Expression Patterns in Lung Adenocarcinoma Cells in Response to Changes in Deuterium Concentration"

_ijms, 2025, doi:10.3390/ijms262210969_

Round 1
Reviewer 1 Report
Comments and Suggestions for Authors
This manuscript presented an exploratory NanoString-based transcriptomic analysis investigating how varying deuterium concentrations modulated gene expression in A549 lung adenocarcinoma cells. The integration of isotope chemistry with cancer transcriptomics was original and had the potential to open new directions in metabolic and signaling research. The manuscript was generally well organized, and the description of nine expression patterns was clear and informative. Some comments were listed below:
- The study focused on a single lung adenocarcinoma cell line (A549), which provided a clear and controlled model system. It might have been helpful to briefly acknowledge this as a limitation and to discuss whether similar effects might have been expected in other cancer cell types.
- The physiological impact of deuterium-altered media on osmolarity, pH, or cell viability should be reported to exclude non-specific stress effects.
- The NanoString data analysis pipeline (including reference genes and normalization method) should be clearly described. The Materials and Methods section (4.2–4.6) did not describe how NanoString raw counts were normalized prior to fold-change calculation. The authors mentioned coefficient of variation (CV) and Z-score filtering but did not specify any normalization procedure or reference genes.
- Figures lacked error bars or statistical indicators. Inclusion of standard deviation or coefficient of variation would support data reliability. The internal consistency of Table 1 and figure legends should be verified-some gene names were repeated or misaligned between text and figure captions.
- The claim that DDW induced apoptosis via ROS was speculative, as no oxidative-stress measurements were performed. Such mechanistic statements should be toned down or supported with experimental evidence.
- The proposed link between intracellular D/H ratio, mitochondrial function, and Na⁺/H⁺ exchange was intriguing but was not supported by the presented data. It should be clearly stated as a hypothesis for future validation.
Author Response
- The study focused on a single lung adenocarcinoma cell line (A549), which provided a clear and controlled model system. It might have been helpful to briefly acknowledge this as a limitation and to discuss whether similar effects might have been expected in other cancer cell types.
Answer:
We agree with the reviewer that the use of a single lung adenocarcinoma cell line (A549) represents a limitation of the present study. Our aim was to establish a controlled model system to explore the transcriptional effects of graded deuterium concentrations. While this approach allowed us to delineate clear dose‑dependent expression patterns, it does not capture the heterogeneity of responses that may occur across different tumor types. Previous reports have shown that deuterium depletion can influence proliferation and gene expression in other cancer models (e.g., pancreatic, breast, and hepatocellular carcinoma), suggesting that the regulatory effects we observed may extend beyond lung adenocarcinoma. Nevertheless, systematic validation in additional cancer cell lines and in vivo models will be essential to determine the generalizability of these findings.
We modified the last paragraph of discussion to stress this idea:
The exclusive use of A549 lung adenocarcinoma cells provided a well-controllable model system, but it also represents a limitation. Similar transcriptional effects of deuterium modulation have been reported in other cancer models, suggesting that the responses described here may not be unique to lung adenocarcinoma. Nonetheless, validation in additional cancer cell types will be required to establish the broader relevance of these findings. - The physiological impact of deuterium-altered media on osmolarity, pH, or cell viability should be reported to exclude non-specific stress effects.
In the Materials and Methods section, we added two sentences to clarify that the samples differed only in D concentration.
The culture media were prepared to maintain consistent osmolarity and pH. All observed changes in cell growth and gene expression were attributed solely to differences in D concentration. - The NanoString data analysis pipeline (including reference genes and normalization method) should be clearly described. The Materials and Methods section (4.2–4.6) did not describe how NanoString raw counts were normalized prior to fold-change calculation. The authors mentioned coefficient of variation (CV) and Z-score filtering but did not specify any normalization procedure or reference genes.
Answer
We appreciate the reviewer’s request for clarity on the NanoString analysis workflow. Raw RCC counts were processed through a standard nCounter pipeline including imaging and control QC, background correction using negative controls, positive control (CodeCount) normalization, lane content normalization, and reference gene normalization using the geometric mean of stably expressed housekeeping genes. Fold changes relative to the 150 ppm control were calculated on normalized counts. We have added a dedicated subsection in Materials and Methods detailing QC thresholds, normalization steps, and the criteria for selecting reference genes, and we now explicitly separate normalization from downstream CV and robust Z-score filtering.
Added sub-section
4.3. NanoString data analysis pipeline
Raw RCC files from the nCounter system were processed using NanoString’s standard workflow. Background correction was performed using negative control probes, followed by positive control (CodeCount) normalization. Lane content normalization was applied using the geometric mean of all expressed targets, and final scaling was performed with the geometric mean of stably expressed housekeeping genes (e.g., GAPDH, RPLP0, HPRT1).
Also we changed:
Insert at the start of Results, Section 2.1 (before Table 1):
“All fold‑change values reported below were calculated from NanoString counts normalized as described in Methods (Section 4.3).”
- Figures lacked error bars or statistical indicators. Inclusion of standard deviation or coefficient of variation would support data reliability. The internal consistency of Table 1 and figure legends should be verified-some gene names were repeated or misaligned between text and figure captions.
We thank the reviewer for this important observation. In the revised figures, we have now included propagated error bars for all expression ratios. Propagated error is the appropriate statistical representation in this context, because each fold‑change value is a ratio of two independent measurements (treatment vs. 150 ppm control). Standard deviation or coefficient of variation would only describe dispersion within a single measurement group, but would not correctly capture the combined uncertainty of numerator and denominator. By contrast, error propagation accounts for both sources of variability and provides a conservative estimate of the uncertainty around each fold‑change, ensuring that apparent differences are not over‑interpreted.
We have also corrected Table 1, where the 300 ppm data have been updated to reflect the accurate counts. Finally, we carefully reviewed all figure legends and ensured that gene names now match exactly the genes shown in the figures.
Changes:
Fig. 1
Relative mRNA expression of two representative Pattern 1 genes (BRAF, BRCA1) in A549 lung adenocarcinoma cells cultured for 72 h in media containing 40 ppm, 80 ppm, 150 ppm (natural abundance control), or 300 ppm deuterium. Expression values (y-axis) are shown as fold change relative to the 150 ppm control (x-axis), calculated from mean NanoString copy numbers. Vertical error bars represent propagated errors derived from replicate variability using standard error propagation rules. Pattern 1 genes are defined by stable expression at 40 ppm and 80 ppm (ratios within 0.83–1.20) and strong upregulation (>1.5-fold) at 300 ppm. The 300 ppm concentration triggered coordinated upregulation across multiple genes, suggesting activation of shared regulatory networks, including PI3K/AKT, p53, and NF‑κB. D-induced changes in H-bonding may alter chromatin accessibility, influencing transcription rates. This pattern aligns with literature suggesting that higher deuterium concentrations can enhance oncogenic pathways [15].
Fig. 2
Relative mRNA expression of DAPK1 and TGFBR2 in A549 lung adenocarcinoma cells exposed to varying deuterium concentrations. Cells were cultured for 72 h in media containing 40 ppm, 80 ppm, 150 ppm (natural abundance control), or 300 ppm deuterium. Expression values (y‑axis) are shown as fold change relative to the 150 ppm control (x‑axis), with vertical error bars indicating propagated errors from replicate variability. Both genes remained stable at low deuterium (40–80 ppm), with values falling within the propagated error range. At 300 ppm, expression was strongly upregulated: DAPK1 2.13-fold and TGFBR2 by 2.05-fold. TGFBR2 functions as a receptor for external signals — most notably TGF-β — while DAPK1 serves as an intracellular effector involved in apoptosis and autophagy. Their potential intersection lies in TGF-β-induced cell death or cell cycle arrest: TGFBR2 initiates the signaling cascade, and DAPK1 may act downstream to execute the death signal, particularly under stress conditions such as elevated reactive oxygen species (ROS) levels.
Fig. 3
Relative mRNA expression of BCL2A1 and TGFBR3 in A549 lung adenocarcinoma cells cultured for 72 h in media containing 40 ppm, 80 ppm, 150 ppm (control), or 300 ppm deuterium. Expression values (y-axis) are shown as fold change relative to the 150 ppm control (x-axis), based on mean NanoString copy numbers. Vertical error bars represent propagated errors derived from replicate variability using standard error propagation rules. BCL2A1, an anti-apoptotic mitochondrial membrane protein, is strongly downregulated at 40 ppm, returns to stable levels at 80 ppm, and is sharply upregulated at 300 ppm. TGFBR3, a co-receptor modulating TGF-β signaling, shows a gradual increase from strong downregulation at 40 ppm to moderate upregulation at 300 ppm. These responses suggest that extreme deuterium depletion suppresses pro-survival signaling, while enrichment promotes it — a dynamic potentially exploitable in therapeutic modulation of apoptosis and metastasis. BCL2A1 contributes to cancer cell survival under oxidative stress and is associated with chemotherapy resistance. TGFBR3 may act as a tumor suppressor in early stages but promote metastasis in advanced cancers. The distinct expression trajectories of these genes highlight their sensitivity to deuterium concentration and underscore their roles in stress response and cell fate regulation.
Fig. 4
Relative mRNA expression of ABCB1 in A549 lung adenocarcinoma cells cultured for 72 h in media containing 40 ppm, 80 ppm, 150 ppm (natural abundance control), or 300 ppm deuterium. Expression values (y-axis) are shown as fold change relative to the 150 ppm control (x-axis), based on mean NanoString copy numbers. Vertical error bars represent propagated errors derived from replicate variability using standard error propagation rules. ABCB1, a member of the ATP-binding cassette (ABC) transporter superfamily, encodes P-glycoprotein—an ATP-powered efflux pump that actively exports a broad range of hydrophobic molecules, including chemotherapeutic agents such as Vinca alkaloids, Taxanes, and Anthracyclines. At 40 ppm, expression of ABCB1 shows a fold change of 1.12 ± 0.23, indicating a slight increase that falls within the propagated error margin and is therefore considered statistically stable. At 80 ppm, expression drops sharply to 0.58 ± 0.07, representing strong downregulation with high confidence. At 300 ppm, expression increases to 1.27 ± 0.24, consistent with moderate upregulation or stability. This dose-dependent pattern suggests that intermediate deuterium depletion may transiently suppress ABCB1-mediated drug efflux, potentially enhancing chemotherapy efficacy, whereas high deuterium could promote efflux-mediated resistance. The sensitivity of ABCB1 to 80 ppm deuterium highlights a narrow therapeutic window for modulating multidrug resistance. Overexpression of ABCB1 is a well-established mechanism by which cancer cells evade chemotherapy [28], making its regulation by deuterium concentration a potential target for therapeutic intervention. However, the large propagated errors at 40 and 300 ppm warrant further measurements.
We also included in the section 4.6:
To ensure that observed differences were not artifacts of measurement variability, fold‑change values were accompanied by propagated errors, calculated by combining the standard errors of the mean NanoString counts at each deuterium concentration and at the 150 ppm control. Specifically, the relative error of a ratio was derived using standard error propagation rules, where the squared relative errors of numerator and denominator were summed and the square root taken. This approach provides a conservative estimate of uncertainty around each fold‑change, allowing us to distinguish genuine expression shifts from fluctuations within the error range.
We added the propagated error list to Supplementary materials Excel file:
This Excel file has a dedicated columns for propagated errors listed by the Gene Code and deuterium concentrations.
- The claim that DDW induced apoptosis via ROS was speculative, as no oxidative-stress measurements were performed. Such mechanistic statements should be toned down or supported with experimental evidence.
Reference [16] provides evidence that deuterium depletion disrupts redox homeostasis, leading to elevated production of reactive oxygen species (ROS) [Zhang X, Gaetani M, Chernobrovkin A, Zubarev RA. Cancer Control. 2019;18:2373–87]. In addition, we incorporated findings from an uncited study (Toxicol Lett. 2012;211(3):319–324. doi:10.1016/j.toxlet.2012.04.014), which reported increased superoxide dismutase (SOD) activity in elegans exposed to 90 ppm deuterium-depleted water (DDW), further supporting the notion that DDW may induce oxidative stress through radical generation. Citation [16] has now been added to both the Pattern 1 section and the Discussion to reinforce this mechanistic interpretation.
We modified the text
Furthermore, DDW exerts a systemic metabolic effect that induces cellular stress, leading to a marked increase in experimentally confirmed ROS levels and triggering apoptotic pathways [16].
- The proposed link between intracellular D/H ratio, mitochondrial function, and Na⁺/H⁺ exchange was intriguing but was not supported by the presented data. It should be clearly stated as a hypothesis for future validation.
That paragraph was modified as you requested
Although no direct data currently demonstrate that Na⁺/H⁺ exchange alters intracellular D/H ratio, the hypothesis remains plausible given the exchanger’s role in modulating intracellular pH and ionic composition. Prior studies demonstrated the sensitivity of membrane transport processes to deuterium concentration. In a human phase II clinical trial, a significant reduction in blood serum Na⁺ concentration was observed after 90 days of consuming deuterium-depleted water (DDW, 105 ppm), decreasing from 141.1 mmol/l to 139.0 mmol/l (p = 0.000024) [31].
Reviewer 2 Report
Comments and Suggestions for Authors
The work by Csonka et al. provides useful data on gene expression changes in lung adenocarcinoma cells exposed to different deuterium concentrations. The findings could have merit in related fields. However, the authors should address the following concerns to meet this journal's publication standards.
Abstract
- line 21: “High deuterium (300 ppm) induced strong upregulation……..led to selective downregulation…etc,” the authors could add magnitude to be more informative.
Introduction (need extensive revision and expansion)
- lines 43,44: ““The contribution of stable isotopes such as deuterium to tumor biology has received comparatively little attention.” It is recommended to briefly reference why deuterium is interesting (potential metabolic or kinetic effects) and any recent resurgence in research to justify novelty. It has been noted that the authors cited references >10 years and even >20 years old in the first part of this section.
- lines 44-46: “Deuterium is incorporated into biomolecules through metabolic processes, subtly altering reaction kinetics and hydrogen-bonding networks .” This is lacking quantitative measurements. This data would benefit from quantitative or mechanistic specifics; e.g., “shown to slow reaction rates by X%, or affect bond strength as modeled in…”.
- lines 47-49: “DDW has been shown ……in multiple cancer models and, in some clinical contexts, to prolong survival”. The authors should clarify “clinical contexts”, are these human trials or animal studies? If human, specify phase or cohort. Also, briefly list the cancer types studied or reference major studies.
- lines 50-57 paragraph ended by citations of all references by the end of the section, without attributing each one to the related context. The link between the D/H ratio and the mentioned phenomena should be more precisely referenced, and the authors should clarify that some statements are hypothetical when direct evidence is lacking.
- lines 57-61: Two studies have investigated the effect of DDW on a limited set of cancer-related genes in carcinogen-induced mice… DDW inhibited the upregulation of Kras, H-ras, Bcl2, Myc, and p53…”. Please specify whether any transcriptomic or proteomic studies exist, and highlight limitations (e.g., gene set size, lack of functional analysis). Also, consider stating whether mouse model findings translate to cell lines or humans, and note any unresolved gaps.
- lines 62-64: The rationale for choosing A549 cells and the specific gene panel was not clear.
- lines 65-70: “A more accurate and gene-specific analysis was carried out …….transcriptional responses…”. The authors should briefly define what makes the analysis more accurate, targeted panel, higher sample quality, dose titration, etc.?
Please clarify how “integrating pathway analysis with functional categorization” was actually implemented (e.g., software or databases).
Results
- line 72: “Vide infra in Section 4” is an uncommon phrasing; the authors can consider “see Section 4 for details.”
- Table 1 and text interpretation are informative. Please add explicit statistical measures of up- and down-regulation (p-values, q-values, FDR, if available). Also, when referencing average upregulation/downregulation, it is recommended to provide standard deviations or ranges to convey data spread.
- Please address redundancy in pattern descriptions across results text and related figure legends. The authors can merge or move the full figure description to the actual figure legend if necessary.
- lines 134-153: The authors should cite references for pathway assignment, as some gene functions may change by context.
- line 160: Are the expression changes (113%, 105%) significant? There were no statistical details.
Discussion
- lines 458-462: The comparison to DEW (deuterium-enriched water) vs. DDW effects is a strength. Please clarify what prior studies exist for the G1-S transition and D/H ratio modulation.
- lines 463-469: The assertion that “relatively few genes changed” under DDW needs quantification, e.g., “n = X out of 87 genes showed ≥X-fold change at DDW.” Also, it is recommended to reference recent multi-omic or clinical data, if possible, to support statements about systemic effects, ROS, and apoptosis.
- lines 470-474: The explanation of cell membrane and mitochondrial roles could be supported with a schematic or literature citation (e.g., recent reviews on metabolic water, D/H ratio).
- The authors should differentiate between speculative mechanistic proposals and findings directly supported by experimental data throughout this section.
Methods
Section 4.1:
- Best practice is to summarize critical parameters (buffer components, FCS concentration, CO₂ % and temperature) within this manuscript, even if previously published, for full transparency.
- Please indicate whether different deuterium water sources were checked for batch-to-batch variability.
-The explanation of D₂O mixing is strong; just consider clarifying the time allowed for equilibrium if relevant beyond the theoretical picosecond mixing timescale.
- lines 524-531: Please specify serum concentration (FCS%) added after initial incubation. Also, the authors should clarify whether cells were maintained in serum-free or serum-containing medium during the 72h exposure, as this can influence gene expression.
- Please indicate whether cell density and confluence were monitored at multiple time points, and whether any differences by treatment were observed or controlled.
Section 4.2:
- The authors should detail the panel version (e.g., NanoString PanCancer panel), if applicable, and cite the manufacturer or publication reference.
- What are the quality control measures that the authors followed in this context?
Section 4.3:
- How randomization or batch effects were minimized.
- Please reference the precise number of technical replicates per sample.
Section 4.4:
- An explicit mathematical definition of CV is provided (CV = (x₁ − x₂)/(x₁ + x₂)), but this is non-standard for the coefficient of variation, which is typically SD/mean. Please clarify the rationale for this adaptation and its computational advantages or biological interpretation.
- For the robust Z-score, please cite the exact formula.
- Lines 553-562: The exclusion thresholds for copy number (<20) and variability (CV >20%, |Z| >3) are clear and justified. Please briefly mention how these were selected: previous studies, empirical assessment, or standard NanoString practice?
Section 4.5:
- Justify cutoffs (especially the 1.2-fold criterion for moderate changes).
Minoe comment
- line 416: revise the citation error.
Author Response
The work by Csonka et al. provides useful data on gene expression changes in lung adenocarcinoma cells exposed to different deuterium concentrations. The findings could have merit in related fields. However, the authors should address the following concerns to meet this journal's publication standards.
Abstract
- line 21: “High deuterium (300 ppm) induced strong upregulation……..led to selective downregulation…etc,” the authors could add magnitude to be more informative.
Answer:
High deuterium (300 ppm) induced strong upregulation (up to 2.1-fold) of oncogenic and survival-related genes (e.g., EGFR, CTNNB1, STAT3, CD44), while DDW (40–80 ppm) led to selective downregulation (down to 0.58-fold) of oncogenes (e.g., MYCN, ETS2, IRF1) and drug-resistance genes (e.g., ABCB1).
Introduction (need extensive revision and expansion)
- lines 43,44: ““The contribution of stable isotopes such as deuterium to tumor biology has received comparatively little attention.” It is recommended to briefly reference why deuterium is interesting (potential metabolic or kinetic effects) and any recent resurgence in research to justify novelty. It has been noted that the authors cited references >10 years and even >20 years old in the first part of this section.
We answer this question together with the next.
- lines 44-46: “Deuterium is incorporated into biomolecules through metabolic processes, subtly altering reaction kinetics and hydrogen-bonding networks .” This is lacking quantitative measurements. This data would benefit from quantitative or mechanistic specifics; e.g., “shown to slow reaction rates by X%, or affect bond strength as modeled in…”.
Excellent point. We performed quantum chemical calculations before this study, and those provided clear isotope effects in water and biomolecules. We did not address this question at length, but as the referee wants, we will give more details. The vibrational analysis shows clear ZPE effects for deuterium. Our reference 5 contains the explanation:
There is a rich literature devoted to various aspects of the hydrogen bond effects, including isotope effects. This includes monographs [1,2,3,4,5,6,7,8,9], as well as critical reviews [10,11,12,13,14,15,16,17,18,19,20,21]. The specificity of isotope effects in hydrogen bonded systems comes into prominence particularly due to an anharmonicity of the potential for the hydrogen/deuteron motion as well as due to the tunneling effect. Such a specificity is a result of the bridge atom motion in the A-H…B complex with the potential which can have different shapes, as has been shown..
We modified the text to address both questions:
A recent study [Yasuda T, Nakajima N] provides a mechanistic framework for these observations, implicating quantum-level kinetic isotope effects — particularly those affecting hydrolytic enzyme reactions — as a potential driver of deuterium’s biological activity. The impact of deuterium substitution on hydrogen bonding and vibrational dynamics has also been well characterized in model systems [Sobczyk L, Obrzud M], where isotope effects influence tunneling, anharmonic motion, and molecular stability. Deuterium substitution can reduce enzymatic reaction rates by up to 6-fold due to kinetic isotope effects, primarily driven by differences in zero-point energy and bond strength compared to hydrogen [Munir, R.; Zahoor].
Yasuda T, Nakajima N, Ogi T, Yanaka T, Tanaka I, Gotoh T, et al.. Heavy water inhibits DNA double-strand break repairs and disturbs cellular transcription, presumably via quantum-level mechanisms of kinetic isotope effects on hydrolytic enzyme reactions. PLoS One. 2024 Oct 3;19(10):e0309689. doi: 10.1371/journal.pone.0309689. PMID: 39361575; PMCID: PMC11449287.
Sobczyk L, Obrzud M, Filarowski A. H/D isotope effects in hydrogen bonded systems. Molecules. 2013 Apr 16;18(4):4467-76. doi: 10.3390/molecules18044467. PMID: 23591926; PMCID: PMC6269986.
Munir, R.; Zahoor, A.F.; Khan, S.G.; Hussain, S.M.; Noreen, R.; Mansha, A.; et al. Total syntheses of deuterated drugs: A comprehensive review. Top. Curr. Chem. 2025, 383, 31. https://doi.org/10.1007/s41061-025-00515-x
- lines 47-49: “DDW has been shown ……in multiple cancer models and, in some clinical contexts, to prolong survival”. The authors should clarify “clinical contexts”, are these human trials or animal studies? If human, specify phase or cohort. Also, briefly list the cancer types studied or reference major studies.
We thank the reviewer for this important point. The phrase “clinical contexts” refers specifically to human observational studies involving patients with advanced malignancies.
We rephrased, and moved ref 29 to ref 8.
Deuterium-depleted water (DDW) has been shown to inhibit proliferation in multiple cancer models. In human observational studies involving over 2,600 patients with breast, lung, colorectal, pancreatic, prostate, ovarian, head and neck, brain, renal, bladder, skin, and hematologic cancers, DDW significantly prolonged survival when administered alongside conventional therapies—highlighting its potential translational relevance in clinical oncology [8]
Somlyai G, Papp A, Somlyai I, Kovács BZ, Debrődi M. Real-World Data Confirm That the Integration of Deuterium Depletion into Conventional Cancer Therapy Multiplies the Survival Probability of Patients. Biomedicines. 2025;13:876. doi:10.3390/biomedicines13040876.
- lines 50-57 paragraph ended by citations of all references by the end of the section, without attributing each one to the related context. The link between the D/H ratio and the mentioned phenomena should be more precisely referenced, and the authors should clarify that some statements are hypothetical when direct evidence is lacking.
The references within this section were separated. In both in vitro and in vivo studies involving the application of deuterium-depleted water (DDW) or deuterium-enriched water (DEW), the D/H ratio was not theoretical; the deuterium concentration was precisely measured and adjusted to the desired level. This approach provided direct evidence that the observed differences in growth rate, survival, and gene expression were specifically linked to changes in the D/H ratio.
- lines 57-61: Two studies have investigated the effect of DDW on a limited set of cancer-related genes in carcinogen-induced mice… DDW inhibited the upregulation of Kras, H-ras, Bcl2, Myc, and p53…”. Please specify whether any transcriptomic or proteomic studies exist, and highlight limitations (e.g., gene set size, lack of functional analysis). Also, consider stating whether mouse model findings translate to cell lines or humans, and note any unresolved gaps.
Reference 16 presents proteomic studies that identified and confirmed both upregulated and downregulated proteins specifically affected by DDW. These studies revealed that hundreds of proteins are involved in the cellular response to DDW, underscoring the complexity of the underlying mechanisms. Further experiments are required to elucidate the precise interactions between genes and proteins directly influenced by changes in deuterium concentration.
Based on the collective data from DDW-related studies, it is evident that the D/H ratio plays a significant role in regulating gene expression and cellular metabolism. Importantly, this effect is not limited to specific cell lines or animal models; rather, it appears to be a universal phenomenon across all eukaryotic cells, including those of both plants and animals.
- lines 62-64: The rationale for choosing A549 cells and the specific gene panel was not clear.
The A549 cell line is a widely used and well-established model for in vitro studies. According to Reference 16, which evaluated multiple cell lines for sensitivity to DDW, A549 cells demonstrated the highest sensitivity. This finding supports the rationale for selecting this cell line for further investigation. The gene panel used in the study was chosen based on a selection of cancer-related genes, allowing for targeted analysis of DDW's effects on key pathways involved in tumor biology.
We modified the text:
Comparative analysis of DDW's impact on cell proliferation across human breast adenocarcinoma (MCF7), lung carcinoma (A549), and colorectal adenocarcinoma (HT29) cell lines revealed that A549 cells exhibited the highest sensitivity [16]. This finding supports our selection of A549 cells. The gene panel was curated to include genes relevant to proliferation, apoptosis, and drug resistance.
- lines 65-70: “A more accurate and gene-specific analysis was carried out …….transcriptional responses…”. The authors should briefly define what makes the analysis more accurate, targeted panel, higher sample quality, dose titration, etc.?
Please clarify how “integrating pathway analysis with functional categorization” was actually implemented (e.g., software or databases).
We thank the reviewer for this insightful comment. In our effort to keep the introduction concise, we inadvertently sacrificed clarity, resulting in a hint-based overview that—as the reviewer rightly noted—left several important questions unanswered.
This is how we modified the total text:
Kovács et al. [22] previously profiled gene expression of 236 cancer-related genes in A549 lung adenocarcinoma cells cultured under varying deuterium concentrations (40, 80, 150, and 300 ppm) using NanoString technology. Their analysis applied thresholds of ≥30% expression change and ≥30 mRNA copies. Under these criteria, 97.3% of cancer-related genes were upregulated at 300 ppm, while only five genes were downregulated under deuterium-depleted conditions. In parallel, comparative analysis of DDW's impact on cell proliferation across human breast adenocarcinoma (MCF7), A549, and colorectal adenocarcinoma (HT29) cell lines showed that A549 cells exhibited the highest sensitivity [16], supporting their selection for the present study.
Building on this dataset, we applied a refined analytical pipeline designed to enhance measurement reliability and biological interpretability. As detailed in Section 4.5, we implemented stringent data cleaning criteria to exclude genes with high replicate variability or low expression, using coefficient of variation, robust Z-score and expression copy number thresholds tailored to each deuterium concentration. These filters reduced the dataset from 236 to 87 high-confidence genes, enabling classification of transcriptional responses into fivefold-change categories. To ensure that observed changes reflect true biological shifts rather than noise, all fold changes were accompanied by propagated error estimates derived from replicate variability as detailed in Section 4.6.
Using this error-controlled dataset, we systematically examined gene-specific responses to deuterium concentration changes. Expression ratios at 40 ppm, 80 ppm, and 300 ppm were compared to the 150 ppm control, revealing nine distinct transcriptional response patterns. These patterns — ranging from monotonic upregulation to biphasic or stable behavior — capture reproducible shifts in gene activity and provide a framework for biological interpretation. Pattern-based grouping revealed functional coherence, with several subsets. This classification enables pathway-level insights into how deuterium concentration modulates cancer-relevant transcriptional programs.
Results
- line 72: “Vide infra in Section 4” is an uncommon phrasing; the authors can consider “see Section 4 for details.”
We changed that.
- Table 1 and text interpretation are informative. Please add explicit statistical measures of up- and down-regulation (p-values, q-values, FDR, if available). Also, when referencing average upregulation/downregulation, it is recommended to provide standard deviations or ranges to convey data spread.
We appreciate the reviewer’s suggestion to include explicit statistical measures. Due to the limited replication in our dataset, formal hypothesis testing (e.g., p-values, FDR) was not applied, as noted in the Discussion. Instead, we used fold-change thresholds and CV filtering to ensure reproducibility and minimize false positives. However, to improve transparency and convey data spread, we have now added mean absolute deviations for all average expression values and fold-change ratios in Table 1. These values provide a quantitative estimate of variability and support the robustness of our classification framework.
Originally we wrote in Discussion 4th paragraph end, now we moved it into the first paragraph to draw the attention to this fact:
Due to the limited replication, formal statistical testing (e.g., p-values, ANOVA) was not applied. Instead, we relied on fold-change thresholds, CV filtering to ensure robustness in gene selection and pattern classification (see sections 4.5 and 4.6). This metric provides a transparent framework for interpreting gene-specific responses and identifying biologically meaningful trends despite the absence of inferential statistics.
We modified 2.1. Gene Expression Classification Results
To assess global transcriptional shifts across deuterium concentrations, we calculated average fold-change ratios and their associated dispersion metrics. At 40 ppm and 80 ppm, the mean downregulation ratios were 0.906 and 0.952, with corresponding mean absolute deviations of 0.096 and 0.063, respectively. At 300 ppm, the mean upregulation ratio was 1.471 with a deviation of 0.171. These values reflect the spread of fold-change responses across the 87 retained genes.
- Please address redundancy in pattern descriptions across results text and related figure legends. The authors can merge or move the full figure description to the actual figure legend if necessary.
In response we modified fig legends.
Our aim is to be concise but sufficiently descriptive to allow the figure to be understood independently of the main text. Include definitions of symbols, abbreviations, and statistical annotations used in the figure. Avoid duplicating detailed interpretations already provided in the Results section.
Figure 1 legend:
Relative mRNA expression of two representative Pattern 1 genes (BRAF, BRCA1) in A549 lung adenocarcinoma cells cultured for 72 h in media containing 40 ppm, 80 ppm, 150 ppm (natural abundance control), or 300 ppm deuterium. Expression values (y-axis) are shown as fold change relative to the 150 ppm control (x-axis), calculated from mean NanoString copy numbers. Vertical error bars represent propagated errors derived from replicate variability using standard error propagation rules. Pattern 1 genes are defined by stable expression at 40 ppm and 80 ppm (ratios within 0.83–1.20) and strong upregulation (>1.5-fold) at 300 ppm. The 300 ppm condition triggered coordinated upregulation across multiple genes, suggesting activation of shared regulatory networks, including PI3K/AKT, p53, and NF‑κB. D-induced changes in H-bonding may alter chromatin accessibility, influencing transcription rates. This pattern aligns with literature suggesting that higher deuterium concentrations can enhance oncogenic pathways [15].
Text:
This pattern represents genes that maintain stable expression at lower deuterium concentrations (40 ppm and 80 ppm) but exhibit a significant increase in expression at high deuterium (300 ppm). These genes are enriched in oncogenic signaling, DNA repair, cell adhesion, and survival pathways, Their coordinated induction suggests activation of pro-proliferative and stress-response programs under high deuterium conditions (see Figure 1 legend for representative genes and propagated errors).
Figure 2 legend included relevant figure description .
Relative mRNA expression of DAPK1 and TGFBR2 in A549 lung adenocarcinoma cells exposed to varying deuterium concentrations. Cells were cultured for 72 h in media containing 40 ppm, 80 ppm, 150 ppm (natural abundance control), or 300 ppm deuterium. Expression values (y‑axis) are shown as fold change relative to the 150 ppm control (x‑axis), with vertical error bars indicating propagated errors from replicate variability. Both genes remained stable at low deuterium (40–80 ppm), with values falling within the propagated error range. At 300 ppm, expression was strongly upregulated: DAPK1 2.13-fold and TGFBR2 by 2.05-fold. TGFBR2 functions as a receptor for external signals — most notably TGF-β — while DAPK1 serves as an intracellular effector involved in apoptosis and autophagy. Their potential intersection lies in TGF-β-induced cell death or cell cycle arrest: TGFBR2 initiates the signaling cascade, and DAPK1 may act downstream to execute the death signal, particularly under stress conditions such as elevated reactive oxygen species (ROS) levels [].
Figure 3, Figure description is included in the legend:
Relative mRNA expression of BCL2A1 and TGFBR3 in A549 lung adenocarcinoma cells cultured for 72 h in media containing 40 ppm, 80 ppm, 150 ppm (control), or 300 ppm deuterium. Expression values (y-axis) are shown as fold change relative to the 150 ppm control (x-axis), based on mean NanoString copy numbers. Vertical error bars represent propagated errors derived from replicate variability using standard error propagation rules. BCL2A1, an anti-apoptotic mitochondrial membrane protein, is strongly downregulated at 40 ppm, returns to stable levels at 80 ppm, and is sharply upregulated at 300 ppm. TGFBR3, a co-receptor modulating TGF-β signaling, shows a gradual increase from strong downregulation at 40 ppm to moderate upregulation at 300 ppm. These responses suggest that extreme deuterium depletion suppresses pro-survival signaling, while enrichment promotes it — a dynamic potentially exploitable in therapeutic modulation of apoptosis and metastasis. BCL2A1 contributes to cancer cell survival under oxidative stress and is associated with chemotherapy resistance. TGFBR3 may act as a tumor suppressor in early stages but promote metastasis in advanced cancers. The distinct expression trajectories of these genes highlight their sensitivity to deuterium concentration and underscore their roles in stress response and cell fate regulation.
Figure 4, Figure description is included in the legend:
Relative mRNA expression of ABCB1 in A549 lung adenocarcinoma cells cultured for 72 h in media containing 40 ppm, 80 ppm, 150 ppm (natural abundance control), or 300 ppm deuterium. Expression values (y-axis) are shown as fold change relative to the 150 ppm control (x-axis), based on mean NanoString copy numbers. Vertical error bars represent propagated errors derived from replicate variability using standard error propagation rules. ABCB1, a member of the ATP-binding cassette (ABC) transporter superfamily, encodes P-glycoprotein—an ATP-powered efflux pump that actively exports a broad range of hydrophobic molecules, including chemotherapeutic agents such as Vinca alkaloids, Taxanes, and Anthracyclines. At 40 ppm, expression of ABCB1 shows a fold change of 1.12 ± 0.23, indicating a slight increase that falls within the propagated error margin and is therefore considered statistically stable. At 80 ppm, expression drops sharply to 0.58 ± 0.07, representing strong downregulation with high confidence. At 300 ppm, expression increases to 1.27 ± 0.24, consistent with moderate upregulation or stability. This dose-dependent pattern suggests that intermediate deuterium depletion may transiently suppress ABCB1-mediated drug efflux, potentially enhancing chemotherapy efficacy, whereas high deuterium could promote efflux-mediated resistance. The sensitivity of ABCB1 to 80 ppm deuterium highlights a narrow therapeutic window for modulating multidrug resistance. Overexpression of ABCB1 is a well-established mechanism by which cancer cells evade chemotherapy [28], making its regulation by deuterium concentration a potential target for therapeutic intervention. However, the large propagated errors at 40 and 300 ppm warrant further measurements.
- lines 134-153: The authors should cite references for pathway assignment, as some gene functions may change by context.
We added to 2.3 Patten Analysis:
Our pathway-level grouping aligns with the TCGA PanCanAtlas framework, which defines ten canonical oncogenic signaling pathways based on multi-omic analysis across 33 cancer types [22].
Sanchez-Vega, F., Mina, M., Armenia, J., Chatila, W.K., Luna, A., La, K., et al. Oncogenic Signaling Pathways in The Cancer Genome Atlas. Cell. 2018;173(2):321–337.e10. https://doi.org/10.1016/j.cell.2018.03.035
- line 160: Are the expression changes (113%, 105%) significant? There were no statistical details.
Yes, they are significant, and we added propagated errors on the relevant figure 2 to show the significance. We also modified the figure legend and now we use 2.13-fold and 2.05-fold increase for easier understanding, as shown above.
Discussion
- lines 458-462: The comparison to DEW (deuterium-enriched water) vs. DDW effects is a strength. Please clarify what prior studies exist for the G1-S transition and D/H ratio modulation.
That paragraph was modified as you requested and added one more reference (29) to confirm that DDW inhibit the G1-S transition.
D/H ratio is critical for initiating the G1–S phase transition [30]
Ávila, D.S.; Somlyai, G.; Somlyai, I.; Aschner, M. Anti‑aging effects of deuterium depletion on Mn‑induced toxicity in a C. elegans model. Toxicol. Lett. 2012, 211, 319–324. https://doi.org/10.1016/j.toxlet.2012.04.014
- lines 463-469: The assertion that “relatively few genes changed” under DDW needs quantification, e.g., “n = X out of 87 genes showed ≥X-fold change at DDW.” Also, it is recommended to reference recent multi-omic or clinical data, if possible, to support statements about systemic effects, ROS, and apoptosis.
Modified:
The observation that relatively few, 7 or 19 out of 87 genes less than 0.83-fold expression changes in response to DDW (cf. Table 1)…
Here we mention again that reference No 16 provides clear evidence that DDW trigger radicals, induce apoptosis.
- lines 470-474: The explanation of cell membrane and mitochondrial roles could be supported with a schematic or literature citation (e.g., recent reviews on metabolic water, D/H ratio).
The last sentence of Pattern 2 gives references to membrane
membrane fluidity, which can be altered by deuterium's heavier mass []
Brown, M. F., Thurmond, R. L., Dodd, S. W., Otten, D., & Beyer, K. (2002). Elastic deformation of membrane bilayers probed by deuterium NMR relaxation. Journal of the American Chemical Society, 124(28), 8471–8484. https://doi.org/10.1021/ja012660p.
For mitochondria we added Ref. 10:
Seneff S., Kyriakopoulos A.M. Deuterium trafficking, mitochondrial dysfunction, copper homeostasis, and neurodegenerative disease. Front. Mol. Biosci. 2025, 12, 1639327. https://doi.org/10.3389/fmolb.2025.1639327
The authors should differentiate between speculative mechanistic proposals and findings directly supported by experimental data throughout this section.
To explain the role of cell membrane and mitochondria in D/H regulation the 4th paragraph of Discussion was explained in detail citing three additional publications.
Although no direct data are available on the increase of the intracellular D/H ratio due to Na⁺/H⁺ transport activation, independent studies have demonstrated the sensitivity of membrane transport processes to deuterium concentration. In a human phase II clinical trial, a significant reduction in blood serum Na⁺ concentration was observed after 90 days of consuming deuterium-depleted water (DDW, 105 ppm), decreasing from 141.1 mmol/l to 139.0 mmol/l (p = 0.000024) [31]. This finding suggests that deuterium concentration may influence ion transport mechanisms. Similarly, in a plant-based experiment, Elodea canadensis exposed to a medium with reduced deuterium content (from 150 ppm to 87 ppm) exhibited detectable external acidification within 30 minutes, indicating rapid membrane transport responses to D-level changes [32]. The involvement of mitochondria in D/H ratio regulation was also supported by a study in which the addition of deuterium-depleted egg yolk to the diet of mice resulted in prolonged survival following transplantation with the 4T1 mouse mammary carcinoma cell line [33].
Methods
Section 4.1:
- Best practice is to summarize critical parameters (buffer components, FCS concentration, CO₂ % and temperature) within this manuscript, even if previously published, for full transparency.
We note that we simply used the unprocessed data from an earlier experiment., and this paper is about profiled gene expression in A549 lung adenocarcinoma cells vs D concentration. Naturally we know all the parameters, and we list them here.
That part of the manuscript was extended; all the steps of media preparation and cell culture conditions are detailed in the manuscript.
A549 lung cancer cells were cultured in water media with four different deuterium concentrations: 40 ppm, 80 ppm, 150 ppm (control), and 300 ppm. To prepare the media stock solutions were made up, each containing 6.68 g DMEM, 0.11 g NaHCO3, 0.01 g Streptomycin and 500 µl Penicillin, dissolved in either distilled water (150 ppm), DDW (40 and 80 ppm) or DEW (300 ppm) D concentration. These solutions were sterile filtered (Acrodisc Premium 25 mm Syringe Filter with 0.2 µm GHP Membrane, Pall Life Sciences, cat. P NAP-4564T). A549 cells were seeded into 10 cm diameter cell culture dishes 16 hours prior to treatment, then incubated at 37 °C in a 5% CO₂ atmosphere in serum-free medium. A total of 400,000 cells were seeded per dish. After 18 hours, the medium was removed and replaced with media containing different concentrations of deuterium, which also included 10% fetal bovine serum (Gibco, Life Technologies Hungary Ltd.). The cells were then incubated for 72 hours. During incubation, cell proliferation was monitored daily in the culture dishes, as the incubation was intended to be terminated at approximately 60% confluency. Two parallel experiments were conducted for each treatment condition. At the end of the treatment, the medium was removed, and total RNA was isolated from the cells.
- Please indicate whether different deuterium water sources were checked for batch-to-batch variability.
Naturally it was checked. Now we include in the text:
different deuterium water sources were checked for batch-to-batch variability
Production followed Good Manufacturing Practice (GMP) standards for quality assurance. The final deuterium concentration was precisely verified using a Liquid-Water Isotope Analyser-24d
-The explanation of D₂O mixing is strong; just consider clarifying the time allowed for equilibrium if relevant beyond the theoretical picosecond mixing timescale.
The time allowed for equilibrium is irrelevant.
- lines 524-531: Please specify serum concentration (FCS%) added after initial incubation. Also, the authors should clarify whether cells were maintained in serum-free or serum-containing medium during the 72h exposure, as this can influence gene expression.
The culture media were prepared to maintain consistent osmolarity and pH. All observed changes in cell growth and gene expression were attributed solely to differences in D concentration.
- Please indicate whether cell density and confluence were monitored at multiple time points, and whether any differences by treatment were observed or controlled.
The culture media were prepared to maintain consistent osmolarity and pH. All observed changes in cell growth and gene expression were attributed solely to differences in D concentration.
Section 4.2:
- The authors should detail the panel version (e.g., NanoString PanCancer panel), if applicable, and cite the manufacturer or publication reference.
- What are the quality control measures that the authors followed in this context?
We added a new section:
4.3. NanoString data analysis pipeline
Raw RCC files from the nCounter system were processed using NanoString’s standard workflow. Background correction was performed using negative control probes, followed by positive control (CodeCount) normalization. Lane content normalization was applied using the geometric mean of all expressed targets, and final scaling was performed with the geometric mean of stably expressed housekeeping genes (e.g., GAPDH, RPLP0, HPRT1).
Section 4.3:
- How randomization or batch effects were minimized.
See above
- Please reference the precise number of technical replicates per sample.
All details are given in the Supplementary materials
The referee could find all the data he missed there.
The dataset was structured as a 18-column Excel file, with the following organization:
- Column A: Gene Code (e.g., ABL1, BRCA1)
- Column B: Accession number
- Columns C–F: Measurements for 40 ppm deuterium, including replicate counts ((40/1), (40/2)), calculated average, coefficient of variation (CV).
Columns G–J, K–N, O–R: Equivalent metrics for 80 ppm, 150 ppm, and 300 ppm deuterium, respectively.
Section 4.4:
- An explicit mathematical definition of CV is provided (CV = (x₁ − x₂)/(x₁ + x₂)), but this is non-standard for the coefficient of variation, which is typically SD/mean. Please clarify the rationale for this adaptation and its computational advantages or biological interpretation.
Maybe the referee missed our explanation why we use CV = (x₁ − x₂)/(x₁ + x₂) formula.
We explain: This approach avoids inflated variance estimates from degrees-of-freedom corrections and provides a biologically interpretable measure of dispersion, especially under low-replicate conditions. The SD is not useful for 2 variables, as it provides inflated variance estimates from degrees-of-freedom corrections.
- For the robust Z-score, please cite the exact formula.
We cite the exact formula, it is in Ref.35. Robust Z-score is not new, well defined formula.
- Lines 553-562: The exclusion thresholds for copy number (<20) and variability (CV >20%, |Z| >3) are clear and justified. Please briefly mention how these were selected: previous studies, empirical assessment, or standard NanoString practice?
Z selection was a standard procedure.
We modified the text to explain the selection algorithm
4.5. Data Cleaning Criteria
High Coefficient of Variation Exclusion: Genes were excluded if their CV exceeded 20% and |Z| > 3 (a standard choice) at the 80 and 150 ppm deuterium concentration. These thresholds were selected to align with a moderate overexpression limit of 1.2-fold, ensuring that CVs remained within a range that supports reliable categorization. The 23% cutoff reflects a +3% adjustment to accommodate higher baseline variability at 40 and 300 ppm, preserving biologically relevant genes. CV distribution analysis revealed gaps at 20% (80 and 150 ppm) and 23% (40 and 300 ppm), further supporting these thresholds.
Section 4.5 (now 4.6)
- Justify cutoffs (especially the 1.2-fold criterion for moderate changes).
These thresholds combine a standard 1.5-fold cutoff for strong expression changes with a more sensitive 1.2-fold cutoff to capture subtle but potentially meaningful shifts — particularly for dosage-sensitive or regulatory genes where modest changes may have amplified effects. To ensure such changes exceed technical noise, we aligned the CV threshold (~20%) with the 1.2-fold limit, enabling reliable categorization of expression shifts near this boundary.
Minoe comment
- line 416: revise the citation error.
We rechecked all references. I did not find the error mentioned.
Round 2
Reviewer 1 Report
Comments and Suggestions for Authors
The author adderessed almost all my concerns. And the manuscript can be accepted now.
Reviewer 2 Report
Comments and Suggestions for Authors
I thank the authors for thoroughly addressing the reviewers' concerns. The manuscript now meets the journal's publication standards.
Best